# Shepard Convolutional Neural Networks

**Jimmy SJ. Ren**\*
SenseTime Group Limited
rensijie@sensetime.com

**Li Xu**
SenseTime Group Limited
xuli@sensetime.com

**Qiong Yan**
SenseTime Group Limited
yanqiong@sensetime.com

**Wenxiu Sun**
SenseTime Group Limited
sunwenxiu@sensetime.com

## Abstract

Deep learning has recently been introduced to the field of low-level computer vision and image processing. Promising results have been obtained in a number of tasks including super-resolution, inpainting, deconvolution, filtering, etc. However, previously adopted neural network approaches such as convolutional neural networks and sparse auto-encoders are inherently with translation invariant operators. We found this property prevents the deep learning approaches from outperforming the state-of-the-art if the task itself requires translation variant interpolation (TVI). In this paper, we draw on Shepard interpolation and design Shepard Convolutional Neural Networks (ShCNN) which efficiently realizes end-to-end trainable TVI operators in the network. We show that by adding only a few feature maps in the new Shepard layers, the network is able to achieve stronger results than a much deeper architecture. Superior performance on both image inpainting and super-resolution is obtained where our system outperforms previous ones while keeping the running time competitive.

## 1   Introduction

In the past a few years, deep learning has been very successful in addressing many aspects of visual perception problems such as image classification, object detection, face recognition [1, 2, 3], to name a few. Inspired by the breakthrough in high-level computer vision, several attempts have been made very recently to apply deep learning methods in low-level vision as well as image processing tasks. Encouraging results has been obtained in a number of tasks including image super-resolution [4], inpainting [5], denosing [6], image deconvolution [7], dirt removal [8], edge-aware filtering [9] etc. Powerful models with multiple layers of nonlinearity such as convolutional neural networks (CNN), sparse auto-encoders, etc. were used in the previous studies. Notwithstanding the rapid progress and promising performance, we notice that the building blocks of these models are inherently translation invariant when applying to images. The property makes the network architecture less efficient in handling translation variant operators, exemplified by the image interpolation operation.

Figure 1 illustrates the problem of image inpainting, a typical translation variant interpolation (TVI) task. The black region in figure 1(a) indicates the missing region where the four selected patches with missing parts are visualized in figure 1(b). The interpolation process for the central pixel in each patch is done by four different weighting functions shown in the bottom of figure 1(b). This process cannot be simply modeled by a single kernel due to the inherent spatially varying property.

In fact, the TVI operations are common in many vision applications. Image super-resolution, which aims to interpolate a high resolution image with a low resolution observation also suffers from the

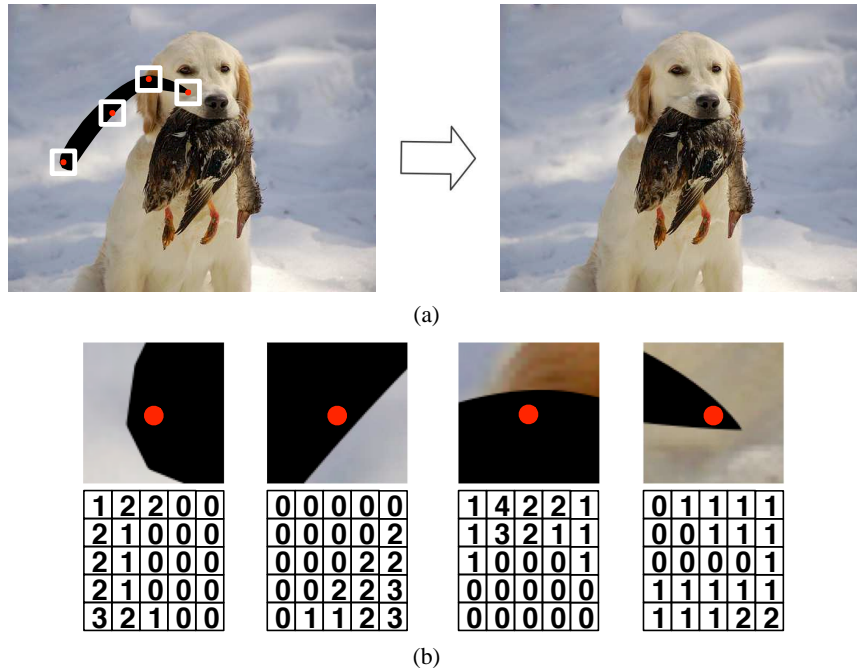

(a)

(b)

Figure 1: Illustration of translation variant interpolation. (a) The application of inpainting. The black regions indicate the missing part. (b) Four selected patches. The bottom row shows the kernels for interpolating the central pixel of each patch.

same problem: different local patches have different pattern of anchor points. We will show that it is thus less optimal to use the traditional convolutional neural network to do the translation variant operations for super-resolution task.

In this paper, we draw on Shepard method [10] and devise a novel CNN architecture named Shepard Convolutional Neural Networks (ShCNN) which efficiently equips conventional CNN with the ability to learn translation variant operations for irregularly spaced data. By adding only a few feature maps in the new Shepard layer and optimizing a more powerful TVI procedure in the end-to-end fashion, the network is able to achieve stronger results than a much deeper architecture. We demonstrate that the resulting system is general enough to benefit a number of applications with TVI operations.

## 2   Related Work

Deep learning methods have recently been introduced to the area of low-level computer vision and image processing. Burger et al. [6] used a simple multi-layer neural network to directly learn a mapping between noisy and clear image patches. Xie et al. [5] adopted a sparse auto-encoder and demonstrated its ability to do blind image inpainting. A three-layer CNN was used in [8] to tackle of problem of rain drop and dirt. It demonstrated the ability of CNN to blindly handle translation variant problem in real world challenges.

Xu et al. [7] advocated the use of generative approaches to guide the design of the CNN for deconvolution tasks. In [9], edge-aware filters can be well approximated using CNN. While it is feasible to use the translation invariant operators, such as convolution, to obtain the translation variant results in a deep neural network architecture, it is less effective in achieving high quality results for interpolation operations. The first attempt using CNN to perform image super-resolution [4] connected the CNN approach to the sparse coding ones. But it failed to beat the state-of-the-art super resolution system [11]. In this paper, we focus on the design of deep neural network layer that better fits the translation variant interpolation tasks. We note that TVI is the essential step for a wide range of

low-level vision applications including inpainting, dirt removal, noise suppression, super-resolution, to name a few.

## 3   Analysis

Deep learning approaches without explicit TVI mechanism generated reasonable results in a few tasks requiring translation variant property. To some extent, deep architecture with multiple layers of nonlinearity is expressive to approximate certain TVI operations given sufficient amount of training data. It is, however, non-trivial to beat non-CNN based approaches while ensuring the high efficiency and simplicity.

To see this, we experimented with the CNN architecture in [4] and [8] and trained a CNN with three convolutional layers by using 1 million synthetic corrupted/clear image pairs. Network and training details as well as the concrete statistics of the data will be covered in the experiment section. Typical test images are shown in the left column of figure 2 whereas the results of this model are displayed in the mid-left column of the same figure. We found that visually very similar results as in [5] are obtained, namely obvious residues of the text are still left in the images. We also experimented with a much deeper network by adding more convolutional layers, virtually replicating the network in [8] by 2,3, and 4 times. Although slight visual differences are found in the results, no fundamental improvement in the missing regions is observed, namely residue still remains.

A sensible next step is to explicitly inform the network about where the missing pixels are so that the network has the opportunity to figure out more plausible solutions for TVI operations. For many applications, the underlying mask indicating the processed regions can be detected or be known in advance. Sample applications include image completion/inpainting, image matting, dirt/impulse noise removal, etc. Other applications such as sparse point propagation and super resolution by nature have the masks for unknown regions.

One way to incorporate the mask into the network is to treat it as an additional channel of the input. We tested this idea with the same set of network and experimental settings as the previous trial. The results showed that such additional piece of information did bring about improvement but still considerably far from satisfactory in removing the residues. Results are visualized in the mid-right column of figure 2. To learn a tractable TVI model, we devise in the next session a novel architecture with an effective mechanism to exploit the information contained in the mask.

## 4   Shepard Convolutional Neural Networks

We initiate the attempt to leverage the traditional interpolation framework to guide the design of neural network architecture for TVI. We turn to the Shepard framework [10] which weighs known pixels differently according to their spatial distances to the processed pixel. Specifically, Shepard method can be re-written in a convolution form

$$J_p = \begin{cases} (\mathbf{K} * I)_p / (\mathbf{K} * \mathbf{M})_p & \text{if} \quad \mathbf{M}_p = 0 \\ I_p & \text{if} \quad \mathbf{M}_p = 1 \end{cases} \tag{1}$$

where $I$ and $J$ are the input and output images, respectively. $p$ indexes the image coordinates. $\mathbf{M}$ is the binary indicator. $\mathbf{M}_p = 0$ indicates the pixel values are unknown. $*$ is the convolution operation. $\mathbf{K}$ is the kernel function with its weights inversely proportional to the distance between a pixel with $\mathbf{M}_p = 1$ and the pixel to process. The element-wise division between the convolved image and the convolved mask naturally controls the way how pixel information is propagated across the regions. It thus enables the capability to handle interpolation for irregularly-spaced data and make it possible translation variant. The key element in Shepard method affecting the interpolation result is the definition of the convolution kernel. We thus propose a new convolutional layer in the light of Shepard method but allow for a more flexible, data-driven kernel design. The layer is referred to as the Shepard interpolation layer.

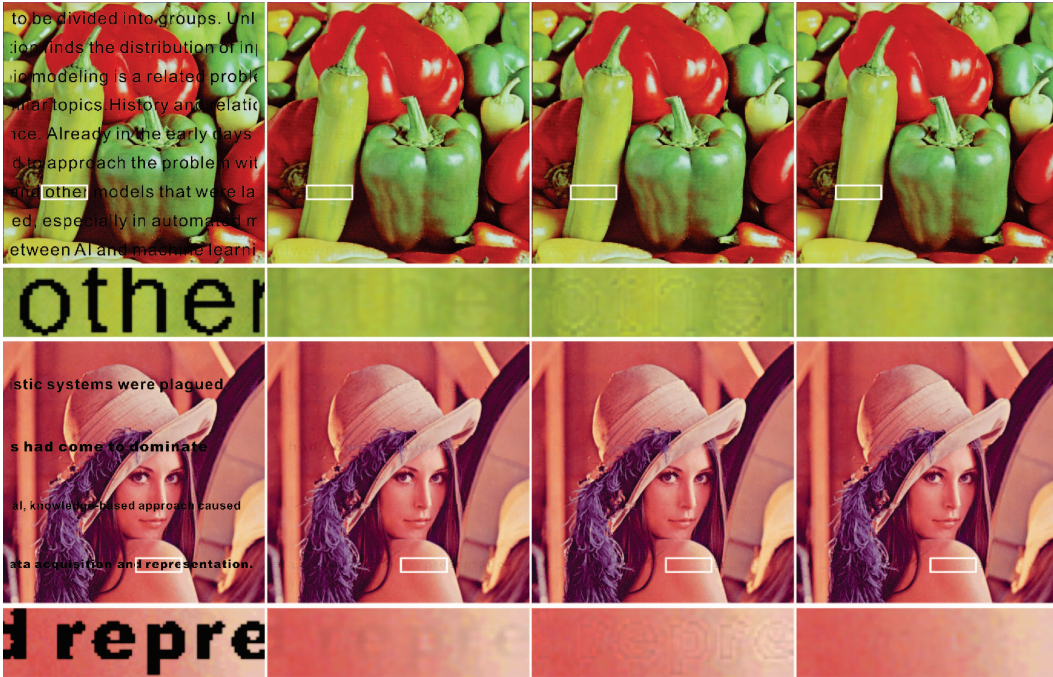

Figure 2: Comparison between ShCNN and CNN in image inpainting. Input images (Left). Results from a regular CNN (Mid-left). Results from a regular CNN trained with masks (Mid-right). Our results (Right).

## 4.1 The Shepard Interpolation Layer

The feed-forward pass of the trainable interpolation layer can be mathematically described as the following equation,

$$\mathcal{F}_i^n(\mathcal{F}^{n-1}, \mathbf{M}^n) = \sigma\left(\sum_j \frac{\mathbf{K}_{ij}^n * \mathcal{F}_j^{n-1}}{\mathbf{K}_{ij}^n * \mathbf{M}_j^n} + b^n\right), \qquad n = 1, 2, 3, \ldots \qquad (2)$$

where $n$ is the index of layers. The subscript $i$ in $\mathcal{F}_i^n$ is the index of feature maps in layer $n$. $j$ in $\mathcal{F}_j^{n-1}$ index the feature maps in layer $n-1$. $\mathcal{F}^{n-1}$ and $\mathbf{M}^n$ are the input and the mask of the current layer respectively. $\mathcal{F}^{n-1}$ represents all the feature maps in layer $n-1$. $\mathbf{K}_{ij}$ are the trainable kernels which are shared in both numerator and denominator in computing the fraction. Concretely, same $\mathbf{K}_{ij}$ is to be convolved with both the activations of the last layer in the numerator and the mask of the current layer $\mathbf{M}^n$ in the denominator. $\mathcal{F}^{n-1}$ could be the output feature maps of regular layers in a CNN such as a convolutional layer or a pooling layer. It could also be a previous Shepard interpolation layer which is a function of both $\mathcal{F}^{n-2}$ and $\mathbf{M}^{n-1}$. Thus Shepard interpolation layers can actually be stacked together to form a highly nonlinear interpolation operator. $b$ is the bias term and $\sigma$ is the nonlinearity imposed to the network. $\mathcal{F}$ is a smooth and differentiable function, therefore standard back-propagation can be used to train the parameters.

Figure 3 illustrates our neural network architecture with Shepard interpolation layers. The inputs of the Shepard interpolation layer are images/feature maps as well as masks indicating where interpolation should occur. Note that the interpolation layer can be applied repeatedly to construct more complex interpolation functions with multiple layers of nonlinearity. The mask is a binary map of value one for the known area, zero for the missing area. Same kernel is applied to the image and the mask. We note that the mask for layer $n+1$ can be automatically generated by the result of previous convolved mask $\mathbf{K}^n * \mathbf{M}^n$, by zeroing out insignificant values and thresholding it. It is important for tasks with relative large missing areas such as inpainting where sophisticated ways of propagation may be learned from data by multi-stage Shepard interpolation layer with nonlinearity. This is also a flexible way to balance the kernel size and the depth of the network. We refer to

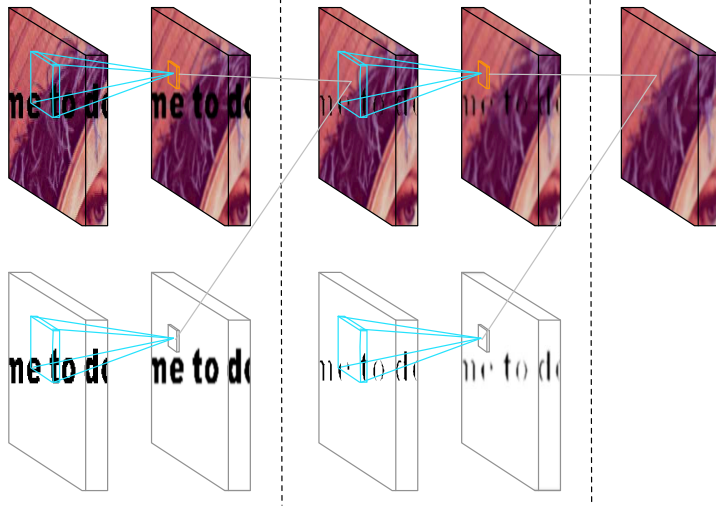

Figure 3: Illustration of ShCNN architecture for multiple layers of interpolation.

a convolutional neural network with Shepard interpolation layers as Shepard convolutional neural network (ShCNN).

## 4.2 Discussion

Although standard back-propagation can be used, because $\mathcal{F}$ is a function of both $\mathbf{K}$s in the fraction, matrix form of the quotient rule for derivatives need to be used in deriving the back-propagation equations of the interpolation layer. To make the implementation efficient, we unroll the two convolution operations $\mathbf{K} * \mathcal{F}$ and $\mathbf{K} * \mathbf{M}$ into two matrix multiplications denoted $\mathbf{W} \cdot \mathcal{I}$ and $\mathbf{W} \cdot \mathcal{M}$ where $\mathcal{I}$ and $\mathcal{M}$ are the unrolled versions of $\mathcal{F}$ and $\mathbf{M}$. $\mathbf{W}$ is the rearrangement of the kernels where each kernel is listed in a single row. $\mathcal{E}$ is the error function to compute the distance between the network output and the ground truth. L2 norm is used to compute this distance. We also denote $\mathcal{Z}^n = \frac{\mathbf{K}^n * \mathcal{F}^{n-1}}{\mathbf{K}^n * \mathbf{M}^n}$. The derivative of the error function $\mathcal{E}$ with respect to $\mathcal{Z}^n$, $\delta^n = \frac{\partial \mathcal{E}}{\partial \mathcal{Z}^n}$, can be computed the same way as in previous CNN papers [12, 1]. Once this value is computed, we show that the derivative of $\mathcal{E}$ with respect to the kernels $\mathbf{W}$ connecting $j^{th}$ node in $(n-1)^{th}$ layer to $i^{th}$ node in $n^{th}$ layer can be computed by,

$$\frac{\partial \mathcal{E}}{\partial \mathbf{W}_{ij}^n} = \sum_m \frac{(\mathbf{W}_{ij}^n \cdot \mathcal{M}_{jm}) \cdot \mathcal{I}_{jm} - (\mathbf{W}_{ij}^n \cdot \mathcal{I}_{jm}) \cdot \mathcal{M}_{jm}}{(\mathbf{W}_{ij}^n \cdot \mathcal{M}_{jm})^2} \cdot \delta_{im}, \tag{3}$$

where $m$ is the column index in $I$, $\mathcal{M}$ and $\delta$.

The denominator of each element in the outer summation in Eq. 3 is different. Therefore, the numerator of each summation element has to be computed separately. While this operation can still be efficiently parallelized by vectorization, it requires significantly more memory and computations than the regular CNNs. Though it brings extra workload in training, the new interpolation layer only adds a fraction of more computation during the test time. We can discern this from Eq. 2, the only added operations are the convolution of the mask with the $K$ and the point-wise division. Because the two convolutions shares the same kernel, it can be efficiently implemented by convolving with samples with the batch size of 2. It thus keeps the computation of Shepard interpolation layer competitive compare to the traditional convolution layer.

We note that it is also natural to integrate the interpolation layer to any previous CNN architecture. This is because the new layer only adds a mask input to the convolutional layer, keeping all other interfaces the same. This layer can also degenerate to a fully connected layer because the unrolled version of Eq. 2 merely contains matrix multiplication in the fraction. Therefore, as long as the TVI operators are necessary in the task, no matter where it is needed in the architecture and the type of layer before or after it, the interpolation layer can be seamlessly plugged in.

Last but not least, the interpolation kernels in the layer is learned from data rather than hand-crafted, therefore it is more flexible and could be more powerful than pre-designed kernels. On the other hand, it is end-to-end trainable so that the learned interpolation operators are embedded in the overall optimization objective of the model.

# 5 Experiments

We conducted experiments on two applications involving TVI: the inpainting and the super-resolution. The training data was generated by randomly sampling 1 million patches from 1000 natural images scraped from Flickr. Grayscale patches of size 48x48 were used for both tasks to facilitate the comparison with previous studies. All PSNR comparison in the experiment is based on grayscale results. Our model can be directly extended to process color images.

## 5.1 Inpainting

The natural images are contaminated by masks containing text of different sizes and fonts as shown in figure 2. We assume the binary masks indicating missing regions are known in advance. The ShCNN for inpainting is consists of five layers, two of which are Shepard interpolation layers. We use ReLU function [1] to impose nonlinearity in all our experiments. 4x4 filters were used in the first Shepard layer to generate 8 feature maps, followed by another Shepard interpolation layer with 4x4 filters. The rest of the ShCNN is conventional CNN architecture. The filters for the third layer is with size 9x9x8, which are use to generate 128 feature maps. 1x1x128 filters are used in the fourth layer. 8x8 filters are used to carry out the reconstruction of image details. Visual results are shown in the last column in figure 2. The results of the comparisons are generated using the architecture in [8]. More examples are provided in the project webpage.

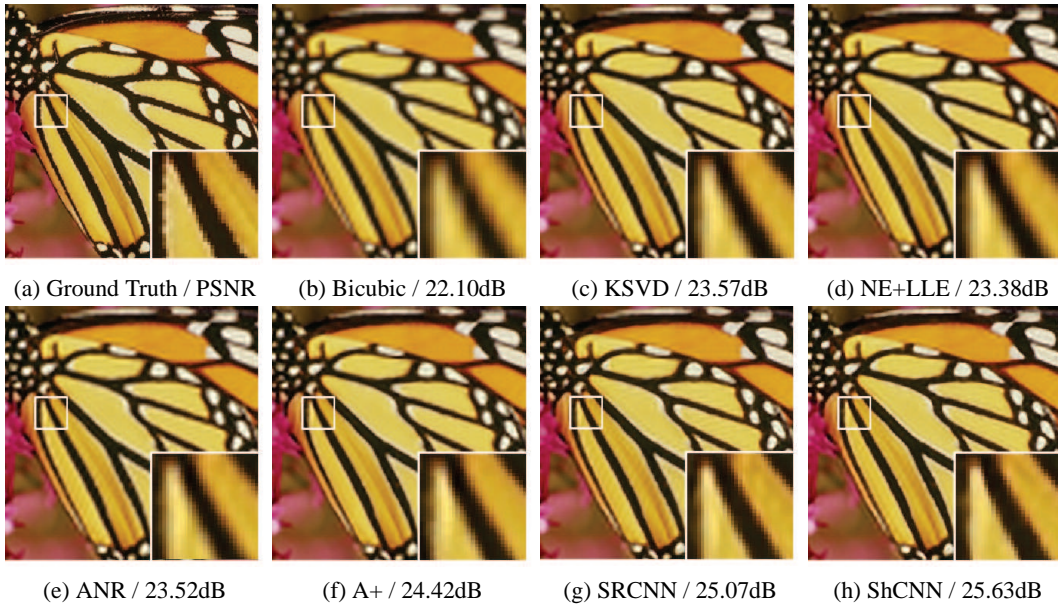

(a) Ground Truth / PSNR     (b) Bicubic / 22.10dB     (c) KSVD / 23.57dB     (d) NE+LLE / 23.38dB

(e) ANR / 23.52dB     (f) A+ / 24.42dB     (g) SRCNN / 25.07dB     (h) ShCNN / 25.63dB

Figure 4: Visual comparison. Factor 4 upscaling of the butterfly image in Set5 [14].

## 5.2 Super Resolution

The quantitative evaluation of super resolution is conducted using synthetic data where the high resolution images are first downscaled by a factor to generate low resolution patches. To perform super resolution, we upscale the low resolution patches and zero out the pixels in the upscaled images, leaving one copy of pixels from low resolution images. In this regard, super resolution can be seemed as a special form of inpainting with repeated patterns of missing area.

| Set14 (x2) | Bicubic | K-SVD | NE+NNLS | NE+LLE | ANR | A+ | SRCNN | ShCNN |
|---|---|---|---|---|---|---|---|---|
| baboon | 24.86dB | 25.47dB | 25.40dB | 25.52dB | 25.54dB | 25.65dB | 25.62dB | **25.79dB** |
| barbara | 28.00dB | 28.70dB | 28.56dB | 28.63dB | 28.59dB | **28.70dB** | 28.59dB | 28.59dB |
| bridge | 26.58dB | 27.55dB | 27.38dB | 27.51dB | 27.54dB | 27.78dB | 27.70dB | **27.92dB** |
| coastguard | 29.12dB | 30.41dB | 30.23dB | 30.38dB | 30.44dB | 30.57dB | 30.49dB | **30.82dB** |
| comic | 26.46dB | 27.89 dB | 27.61dB | 27.72dB | 27.80dB | 28.65dB | 28.27dB | **28.70dB** |
| face | 34.83dB | 35.57 dB | 35.46dB | 35.61dB | 35.63dB | 35.74dB | 35.61dB | **35.75**dB |
| flowers | 30.37dB | 32.28 dB | 31.93dB | 32.19dB | 32.29dB | 33.02dB | 33.03dB | **33.53dB** |
| foreman | 34.14dB | 36.18 dB | 35.93dB | 36.41dB | 36.40dB | **36.94dB** | 36.20dB | 36.14dB |
| lenna | 34.70dB | 36.21 dB | 36.00dB | 36.30dB | 36.32dB | 36.60dB | 36.50dB | **36.71dB** |
| man | 29.25dB | 30.44 dB | 30.29dB | 30.43dB | 30.47dB | 30.87dB | 30.82dB | **31.06dB** |
| monarch | 32.94dB | 35.75 dB | 35.26dB | 35.58dB | 35.71dB | 37.01dB | 37.18dB | **38.09dB** |
| pepper | 34.97dB | 36.59 dB | 36.18dB | 36.36dB | 36.39dB | 37.02dB | 36.75dB | **37.03dB** |
| ppt3 | 26.87dB | 29.30 dB | 28.98dB | 28.97dB | 28.97dB | 30.09dB | 30.40dB | **31.07**dB |
| zebra | 30.63dB | 33.21dB | 32.59dB | 33.00dB | 33.07dB | **33.59dB** | 33.29dB | 33.51dB |
| Avg PSNR | 30.23dB | 31.81dB | 31.55dB | 31.76dB | 31.80dB | 32.28dB | 32.18dB | **32.48dB** |
| Set14 (x3) | Bicubic | K-SVD | NE+NNLS | NE+LLE | ANR | A+ | SRCNN | ShCNN |
| baboon | 23.21dB | 23.52dB | 23.49dB | 23.55dB | 23.56dB | 23.62dB | 23.60dB | **23.69dB** |
| barbara | 26.25dB | 26.76dB | 26.67dB | 26.74dB | 26.69dB | 26.47dB | **26.66dB** | 26.54dB |
| bridge | 24.40dB | 25.02dB | 24.86dB | 24.98dB | 25.01dB | 25.17dB | 25.07dB | **25.28dB** |
| coastguard | 26.55dB | 27.15dB | 27.00dB | 27.07dB | 27.08dB | 27.27dB | 27.20dB | **27.43dB** |
| comic | 23.12dB | 23.96dB | 23.83dB | 23.98dB | 24.04dB | 24.38dB | 24.39dB | **24.70dB** |
| face | 32.82dB | 33.53dB | 33.45dB | 33.56dB | 33.62dB | **33.76dB** | 33.58dB | 33.71dB |
| flowers | 27.23dB | 28.43dB | 28.21dB | 28.38dB | 28.49dB | 29.05dB | 28.97dB | **29.42dB** |
| foreman | 31.18dB | 33.19dB | 32.87dB | 33.21dB | 33.23dB | 34.30dB | 33.35dB | **34.45dB** |
| lenna | 31.68dB | 33.00dB | 32.82dB | 33.01dB | 33.08dB | 33.52dB | 33.39dB | **33.68dB** |
| man | 27.01dB | 27.90dB | 27.72dB | 27.87dB | 27.92dB | 28.28dB | 28.18dB | **28.41dB** |
| monarch | 29.43dB | 31.10dB | 30.76dB | 30.95dB | 31.09dB | 32.14dB | 32.39dB | **33.37dB** |
| pepper | 32.39dB | 34.07dB | 33.56dB | 33.80dB | 33.82dB | 34.74dB | 34.35dB | **34.77dB** |
| ppt3 | 23.71dB | 25.23dB | 24.81dB | 24.94dB | 25.03dB | 26.09dB | 26.02dB | **26.89dB** |
| zebra | 26.63dB | 28.49dB | 28.12dB | 28.31dB | 28.43dB | 28.98dB | 28.87dB | **29.10dB** |
| Avg PSNR | 27.54dB | 28.67dB | 28.44dB | 28.60dB | 28.65dB | 29.13dB | 29.00dB | **29.39dB** |
| Set14 (x4) | Bicubic | K-SVD | NE+NNLS | NE+LLE | ANR | A+ | SRCNN | ShCNN |
| baboon | 22.44dB | 22.66dB | 22.63dB | 22.67dB | 22.69dB | 22.74dB | 22.70dB | **22.75dB** |
| barbara | 25.15dB | 25.58dB | 25.53dB | 25.58dB | 25.60dB | 25.74dB | 25.70dB | **25.80dB** |
| bridge | 23.15dB | 23.65dB | 23.54dB | 23.60dB | 23.63dB | 23.77dB | 23.66dB | **23.83dB** |
| coastguard | 25.48dB | 25.81dB | 25.82dB | 25.81dB | 25.80dB | 25.98dB | 25.93dB | **26.13dB** |
| comic | 21.69dB | 22.31dB | 22.19dB | 22.26dB | 22.33dB | 22.59dB | 22.53dB | **22.74**dB |
| face | 31.55dB | 32.18dB | 32.09dB | 32.19dB | 32.23dB | **32.44dB** | 32.12dB | 32.35dB |
| flowers | 25.52dB | 26.44dB | 26.28dB | 26.38dB | 26.47dB | 26.90dB | 26.84dB | **27.18dB** |
| foreman | 29.41dB | 31.01dB | 30.90dB | 30.90dB | 30.83dB | 32.24dB | 31.47dB | **32.30dB** |
| lenna | 29.84dB | 30.92dB | 30.82dB | 30.93dB | 30.99dB | 31.41dB | 31.20dB | **31.45dB** |
| man | 25.70dB | 26.46dB | 26.30dB | 26.38dB | 26.43dB | 26.78dB | 26.65dB | **26.82dB** |
| monarch | 27.46dB | 28.72dB | 28.48dB | 28.58dB | 28.70dB | 29.39dB | 29.89dB | **30.30dB** |
| pepper | 30.60dB | 32.13dB | 31.78dB | 31.87dB | 31.93dB | **32.87dB** | 32.34dB | 32.82dB |
| ppt3 | 21.98dB | 23.05dB | 22.61dB | 22.77dB | 22.85dB | 23.64dB | 23.84dB | **24.49dB** |
| zebra | 24.08dB | 25.47dB | 25.17dB | 25.36dB | 25.47dB | 25.94dB | 25.97dB | **26.21dB** |
| Avg PSNR | 26.00dB | 26.88dB | 26.72dB | 26.81dB | 26.85dB | 27.32dB | 27.20dB | **27.51dB** |

Table 1: PSNR comparison on the Set14 [13] image set for upscaling of factor 2, 3 and 4. Methods compared: Bicubic, K-SVD [13], NE+NNLS [14], NE+LLE [15], ANR [16], A+ [11], SRCNN [4], Our ShCNN

We use one Shepard interpolation layer at the top with kernel size of 8x8 and feature map number 16. Other configuration of the network is the same as that in our new network for inpainting. During training, weights were randomly initialized by drawing from a Gaussian distribution with zero mean and standard deviation of 0.03. AdaGrad [17] was used in all experiments with learning rate of 0.001 and fudge factor of 1e-6. Table 1 show the quantitative results of our ShCNN in a widely used super-resolution data set [13] for upscaling images 2 times, 3 times and 4 times respectively. We compared our method with 7 methods including the two current state-of-the-art systems [11, 4]. Clear improvement over the state-of-the-art systems can be observed. Visual comparison between our method and the previous methods is illustrated in figure 4 and figure 5.

## 6 Conclusions

In this paper, we disclosed the limitation of previous CNN architectures in image processing tasks in need of translation variant interpolation. New architecture based on Shepard interpolation was proposed and successfully applied to image inpainting and super-resolution. The effectiveness of

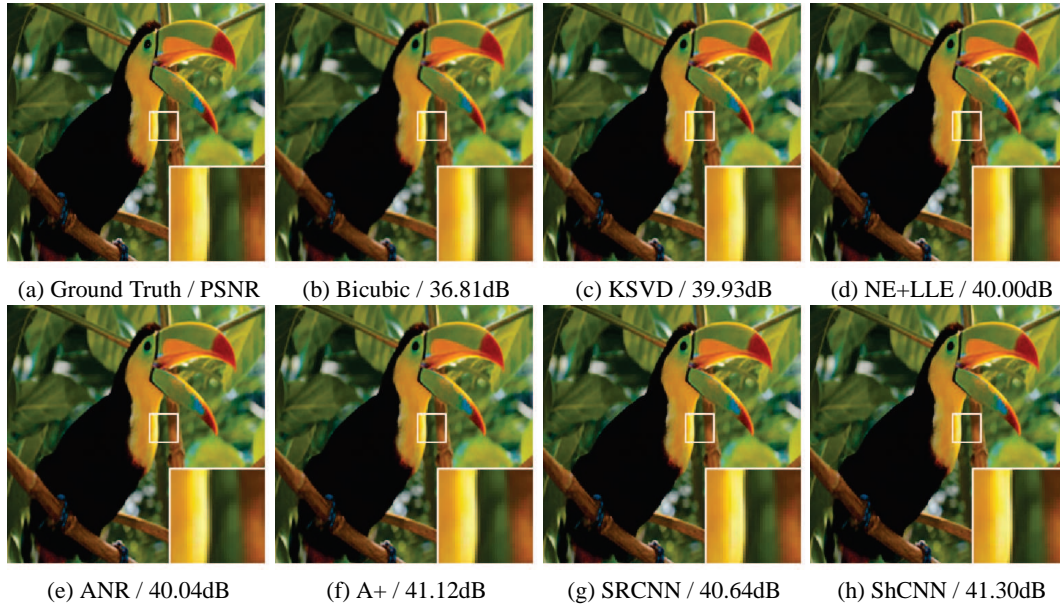

| | | | |
|---|---|---|---|
| (a) Ground Truth / PSNR | (b) Bicubic / 36.81dB | (c) KSVD / 39.93dB | (d) NE+LLE / 40.00dB |
| (e) ANR / 40.04dB | (f) A+ / 41.12dB | (g) SRCNN / 40.64dB | (h) ShCNN / 41.30dB |

Figure 5: Visual comparison. Factor 2 upscaling of the bird image in Set5 [14].

the ShCNN with Shepard interpolation layers have been demonstrated by the state-of-the-art performance.

## Footnotes

\*Project page: http://www.deeplearning.cc/shepardcnn

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
