[Reviews · NeurIPS 2015]

Submitted by Assigned_Reviewer_1

The presentation of this paper needs to be improved.
Summary: This paper adds "Shepard interpolation layer" to low-level image processing CNN to improve performance of inpainting or super-resolution. Scattered interpolation methods are common techniques in interpolating image data. It might be new to combine this with CNN.

Submitted by Assigned_Reviewer_2

PAPER SUMMARY

A recent work [4] proposed using a deep CNN for super-resolution.

Although it generated impressive results, it could not outperform a state-of-the-art super-resolution method based on sparse coding [10].

The authors of this paper argue that this limitation is attributed to the inability of the CNN model in [4] to handle translation variant interpolation (TVI).

To address this problem, they propose using an old interpolation method to develop a CNN variant called Shepard convolutional neural network (ShCNN).

ShCNN is compared with other methods on inpainting and super-resolution tasks.

CRITERION 1: QUALITY

It is widely perceived by many researchers that the translation-invariant property of conventional CNN, though promising for tasks such as robust object recognition, may not be very suitable for many other computer vision and image processing tasks.

For example, in the context of visual tracking (not even a low-level image processing task), it was pointed out by Wang and Yeung in "Learning a deep compact image representation for visual tracking" (NIPS 2013) that a translation-variant CNN is needed for tracking applications.

Likewise, it is reasonable to put forward the working hypothesis that introducing TVI to CNN would help to improve the results for inpainting and super-resolution tasks.

Nevertheless, the authors have not provided justification on why Shepard interpolation is adopted here.

Shepard interpolation is essentially a kernel that imposes an inverse distance weighting function.

What is so special about Shepard interpolation that it is chosen here instead of many other kernel choices?

Does it have computational advantages, or representational advantages, or both?

Justification is missing in the paper.

While it is reasonable to use masks for the inpainting task, is super-resolution with masks also a reasonable problem setting?

The proposed method is evaluated empirically using a quantitative performance measure (PSNR).

There is no discussion on the weaknesses of the proposed method or even analysis of some failure cases.

Since ShCNN adds one or two interpolation layers on top of the original CNN, a more fair comparison is to add the same number of conventional convolutional layers.

CRITERION 2: CLARITY

Although the proposed extension to CNN is relatively simple, the paper presentation falls short of describing the proposed changes clearly and precisely.

Since the paper has not reached the page limit of 9 pages, there should be plenty of space to provide more details about the model and algorithm.

For example, exactly how the convolution form can be rewritten equivalently based on matrix multiplications using the unrolled matrices should be

provided in greater depth in Section 4.1.

How can weight sharing be achieved using the equivalent scheme?

The data preprocessing procedure in the experiments is also confusing.

It seems that some pixels of the high-resolution images are kept unchanged when generating the low-resolution images and they are used to specify the masks for ShCNN.

If this is the case, it may raise a fairness concern as the other methods compared (e.g., [4]) do not use any mask information.

With the mask information, ShCNN somehow already knows part of the "ground truth" of the super-resolved images.

Confusion is also caused by some problems with the paper writing and formatting.

For example, on line #071: "... learn both translational variant operations ..." Since "both" refers to two things, what is the second thing besides "translational variant operations"?

Or do you mean such operations for inpainting and super-resolution?

Also, in the first paragraph on page 4, is there any difference between M and \mathbf{M}?

Or is it simply a formatting error?

It is very confusing to use both regular and bold fonts for matrices (e.g., I and \mathbf{K}).

Moreover, sometimes it is switched between the two font styles (e.g., \mathbf{K} and K (line #240)).

On the last line of page 4, what is UM?

Also, what are the parameters needed for specifying the kernel \mathbf{K}?

As said in the beginning of Section 5, experiments have been conducted on both inpainting and super-resolution tasks.

However, the subsequent results presented seem to be for super-resolution only.

To make it less cluttered, the unit "dB" can be left out in the table entries but only mentioned (once) in the figure caption.

Besides, the writing has much room for improvement.

Here are just a few of the many examples of language errors: #029: "outperfromed" should be written as "outperformed" #035: "In the past a few years" should be written as "In the past few years" #039: "Encouraging results has" should be written as "Encouraging results have" #044: "when applying to" should be written as "when applied to" and many more (particularly in later sections).

There are also some formatting problems.

For example, there should be a space before a citation, i.e., "in [8]" but not "in[8]".

This problem happens a few times.

CRITERION 3: ORIGINALITY

Shepard interpolation is an old method proposed in 1968.

Since the optimization variables are in both the numerator and denominator of the fraction in Eq (1), a more sophisticated learning algorithm (as opposed to applying brute-force back-propagation) would help to handle this highly nonlinear optimization problem better.

Unfortunately such an exploration is missing.

In its current form, the originality of this paper appears minimal.

On the problem of the previous methods discussed in the last paragraph of Section 3.1 and illustrated in Figure 2, actually it is discussed in [5] that K-SVD can already solve this problem.

It should be mentioned in this paper.

The difference is mainly between blind image inpainting and non-blind image inpainting.

In the context of existing work, it weakens the claimed contribution of this paper.

CRITERION 4: SIGNIFICANCE

The paper, especially its early parts, makes some claims that seem a bit too general.

Is the proposed CNN variant for all TVI operations?

Since experimental evaluation has only been done on super-resolution, it is probably more appropriate to make the claim for this particular task only.

Super-resolution is always difficult to evaluate.

It has been found and reported by researchers before that a method that gives higher PSNR may not give visually better result as judged by humans.

One example is that if a method focuses only on the edges (primal sketch) by increasing their resolution, the super-resolved images may look better than those obtained by another method giving higher PSNR.

For a new super-resolution method to bring about significant contribution, it is worth supplementing quantitative comparison with qualitative comparison involving humans.

This possibility is now more feasible with the emergence of crowdsourcing platforms.
Summary: This paper proposes introducing Shepard interpolation to the convolutional neural network (CNN) model to make it exhibit translational variance needed for some image processing tasks.

The ideas are not particularly original and the proposed changes are not technically challenging to realize.

The paper can be much improved in its clarity and linguistic quality.

Submitted by Assigned_Reviewer_3

The paper describes an alternative to a convolutional net layer, in which the weights are determined by an analogy to Shepards interpolation. The latter is a simple and classic interpolation scheme in which a particular point on a function is reconstructed with weights that are inversely proportional to the known data, together with a sum-to-one constraint. The paper shows that by including these "shepard" layers a net can produce image inpainting and superresolution of better quality, and with fewer layers, than using convnets. The idea is original, but is introduced in a somewhat ad hoc way. It is not clear if it is applicable to problems other than images.

The description around line 197 is a bit vague. I also did not understand the paragraph that ends around line 283. There are minor English usage problems, however they mostly do not detract from readability. What is the meaning of the red boxes in Figure 1? Line 266 "groud" (spelling)
Summary: I feel that this is useful work and deserves to be published. It may be more appropriate for a vision or image processing conference however.

Submitted by Assigned_Reviewer_4

This paper proposes a compelling approach to utilizing Shepard (Translation variant interpolation) inside a deep neural network in order to address inpainting and super-resolution tasks. In addition to the TVI layers, the paper utilizes extra layers to signify where the interpolation should occur.

This paper is clearly written and has well-motivated technical content. Although the idea is somewhat straightforward, it is well compensated by strong experimental evidence due to the good execution of technical details. The paper addresses fundamental image processing tasks with good results so it is of interest for the computer community.

Update: after reading the other reviews, I still think that this paper has a lot of merits, but a better outlet for this work would be a more vision oriented conference.

Also I agree with some of the other reviewers that the interpretation of the results requires more thorough analyses and explanations than what was provided in the paper.

I modifying my score down from 7 to 6 to indicate these opinions.

Summary: This paper proposes a well motivated elegant approach to integate the Shepard image interpolation method into an end-to-end trained

deep learning framework. The approach is tested on standard

inpainting and superresolution benchmarks with impressive

qualitative and quantitative results.

Submitted by Assigned_Reviewer_5

# Positive - Combining shepard interpolation [9] with CNNs is an interesting and presumably novel idea. - The paper compares against recent state-of-the-art methods. - The experimental results are good, both visually and quantitatively.

# Negative The exposition of the proposed method is very unclear to me, it needs major improvements. Many parts of the paper are problematic:

- The concept of "translation variant interpolation" (TVI) is only vaguely explained, in part through the example in Fig. 1. However, the concept is never explained more formally. - The discussion of why current models are bad at TVI is unclear and thus not convincing. I understand that the paper intends to show in Sec. 3.1 that current network architectures are inherently unsuited for TVI. However, the paper didn't succeed in convincing me of this at all, because the explanation is simply unclear. Hence, I cannot agree with the sentence in l. 145

("We believe we may conclude from the previous experimental analysis that to learn a tractable TVI model, a novel architecture with an effective mechanism to exploit the information contained in the mask has to be devised.") - The very high-level explanation of shepard interpolation at the bottom of page 3 is not sufficient. Please explain the concept in more detail before delving into the details of the interpolation layer, which is based on this. - There is no single clear explanation of the proposed model architecture. The different components are merely explained in different parts of the paper. Furthermore, Figure 3 is not very informative on its own, since the different layers are not labeled. - In general, the paper quickly delves into details, but fails to give big picture explanations. Page 4 is a prime example of this: It first gives the details of the proposed interpolation layer without first explaining what shepard interpolation is. The following detailed discussion of how gradients can be computed for the new layer is an implementation detail (note, it can even be done automatically via automatic differentiation, cf. http://www.autodiff.org). Finally, it talks about a cost function which presumably is the loss layer of the CNN. The problem is that the entire network architecture has never been explained at that point; only later (page 6) do we learn that the new interpolation layers sit right at the top before the loss layer.

# Miscellaneous - There are many typos in the paper, even in the abstract. The paper needs copy-editing. - The bibliography includes only few references. - Discussion of the analogy (l. 090) between CNN and "multi-layer fully connected (with optional pooling) network" not clear to me. Also, please include a reference for "Caffe" (footnote 1 on page 2).
Summary: Based on shepard interpolation, the paper proposes a novel CNN architecture to be used for image inpainting and super-resolution. Although the experimental results are very good, the explanation of the proposed model needs major improvements.

Author Feedback
Author rebuttal: We thank all the reviewers for the careful and insightful reviews.

R1 has concerns on the originality and significance. We clarify with respect that we made the first attempt to blend Shepard interpolation into an end-to-end trainable CNN framework to perform translation variant interpolation for image processing tasks. A CNN with multiple Shepard layers were shown to have potentials to address challenging inpainting and super-resolution problems, which otherwise cannot be easily addressed using conventional framework. We plan to share our source code with the community after the paper is published. The suggestions on the exposition are readily to be incorporated in the revision.

Other questions

- Mask for super-resolution (R1)

The pipeline of the neural network approach to this task is 1. interpolation; 2. refinement. In [4], the authors used bicubic interpolation as a pre-processing step, so CNN was only used for refinement. By using the Shepard layer, our approach does not assume a pre-defined interpolation kernel. Because our system is trained end-to-end, it is able to learn the interpolation kernels work best with the latter refinement stage network.

- Measurement of super-resolution (R1)

PSNR is one of the most important indicators for measurement. We also provided visual comparisons with many state-of-the-art methods in the paper. The advantage of our method is significant (both quantitatively and visually) for images with complex textures (as in figure 4). For more plain images where bicubic kernel already does a good job, the advantage of using trained interpolation is not obvious.

- Fairness of comparison (R1)

In the analysis section, we pointed out that we compared a CNN with Shepard layers and a CNN with much deeper architecture without Shepard layers (but with mask as an additional input channel). We showed that our approach worked much better (figure 2). For the super-resolution task, the bicubic interpolation in [4] can be thought of a hard-coded layer. So compare it with our network, which only adds one Shepard layer at the top is fair. More importantly, our network also outperformed state-of-the-art sparse coding approach [10] which already performed better than the previous CNN approach in [4].

- Data pre-processing (R1)

To generate the low resolution training samples, the high resolution patches were firstly downscaled by using bicubic interpolation. So, no pixels in the high resolution images were unfairly kept in training. The difference here is we only use nearest neighbor interpolation for upscaling lower resolution input and rely on the new Shepard layer to learn the interpolation kernel rather than merely using bicubic.

- Originality(R1)

It is true that K-SVD addresses similar problem, however, this work is the first to use neural network approach to explicitly perform translation variant interpolation. The adoption of Shepard layers to achieve this has not been done by others. Our approach significantly outperformed the previous K-SVD method both quantitatively and visually in super-resolution.

- Presentation (R3, R4)

Thank you. We note that Shepard interpolation is one efficient way for implementing translation variant interpolation in a parametric form. We will make the concept of TVI and Shepard CNN clearer in the revision.